# Dynamic Prediction of Mechanical Thrombectomy Outcome for Acute Ischemic Stroke Patients Using Machine Learning

**DOI:** 10.3390/brainsci12070938

**Published:** 2022-07-18

**Authors:** Yixing Hu, Tongtong Yang, Juan Zhang, Xixi Wang, Xiaoli Cui, Nihong Chen, Junshan Zhou, Fuping Jiang, Junrong Zhu, Jianjun Zou

**Affiliations:** 1School of Basic Medicine and Clinical Pharmacy, China Pharmaceutical University, Nanjing 210009, China; 3320092130@stu.cpu.edu.cn (Y.H.); 3320092195@stu.cpu.edu.cn (T.Y.); 2Department of Pharmacy, Nanjing First Hospital, Nanjing Medical University, Nanjing 210029, China; junrong_zhu@aliyun.com; 3Department of Neurology, Yuhua Branch of Nanjing First Hospital, Nanjing Yuhua Hospital, Nanjing Medical University, Nanjing 210029, China; zhangjuan_1@126.com (J.Z.); 17720809749@163.com (X.C.); 4Department of Neurology, Nanjing First Hospital, Nanjing Medical University, Nanjing 210029, China; w15895820585@163.com (X.W.); y19850867562@163.com (N.C.); zhjsh333@126.com (J.Z.); 5Department of Geriatrics, Nanjing First Hospital, Nanjing Medical University, Nanjing 210029, China; fupingjiang@163.com; 6Department of Pharmacy, Nanjing First Hospital, China Pharmaceutical University, Nanjing 210029, China

**Keywords:** mechanical thrombectomy, acute ischemic stroke, dynamic prediction, machine learning, unfavorable outcome

## Abstract

The unfavorable outcome of acute ischemic stroke (AIS) with large vessel occlusion (LVO) is related to clinical factors at multiple time points. However, predictive models used for dynamically predicting unfavorable outcomes using clinically relevant preoperative and postoperative time point variables have not been developed. Our goal was to develop a machine learning (ML) model for the dynamic prediction of unfavorable outcomes. We retrospectively reviewed patients with AIS who underwent a consecutive mechanical thrombectomy (MT) from three centers in China between January 2014 and December 2018. Based on the eXtreme gradient boosting (XGBoost) algorithm, we used clinical characteristics on admission (“Admission” Model) and additional variables regarding intraoperative management and the postoperative National Institute of Health stroke scale (NIHSS) score (“24-Hour” Model, “3-Day” Model and “Discharge” Model). The outcome was an unfavorable outcome at the three-month mark (modified Rankin scale, mRS 3–6: unfavorable). The area under the receiver operating characteristic curve and Brier scores were the main evaluating indexes. The unfavorable outcome at the three-month mark was observed in 156 (62.0%) of 238 patients. These four models had a high accuracy in the range of 75.0% to 87.5% and had a good discrimination with AUC in the range of 0.824 to 0.945 on the testing set. The Brier scores of the four models ranged from 0.122 to 0.083 and showed a good predictive ability on the testing set. This is the first dynamic, preoperative and postoperative predictive model constructed for AIS patients who underwent MT, which is more accurate than the previous prediction model. The preoperative model could be used to predict the clinical outcome before MT and support the decision to perform MT, and the postoperative models would further improve the predictive accuracy of the clinical outcome after MT and timely adjust therapeutic strategies.

## 1. Introduction

Acute ischemic stroke (AIS) is the main cause of lifelong disability and mortality around the world [1,2]. Therefore, mechanical thrombectomy (MT) is now regarded as a standard of care for managing patients with acute large vessel occlusion stroke [3,4,5,6]. However, despite high recanalization rates, almost one-third of stroke patients show unfavorable outcomes [7,8]. Hence, the early dynamic prediction of an unfavorable outcome is important in AIS patients treated with MT, and it can be assessed by the modified Rankin Scale (mRS) score [9]. 

Previous studies show that many prognostic factors are associated with the unfavorable outcome in AIS patients, such as physiologic factors [10,11,12] (e.g., age, sex, comorbidity), clinical factors [13,14] (e.g., NHISS score, glycosylated hemoglobin, creatinine), and neuroimaging prognostic factors [15,16,17] (e.g., location of the occlusion, Alberta Stroke Program Early CT Score (ASPECTS)). However, these pretreatment prognostic factors still cannot well predict the outcome of AIS patients. This is due to some interventional factors that also affect the prognosis, such as procedure time, surgical technique, recanalization status, and so on [10,14,18].

Models used for dynamically predicting three-month unfavorable outcomes in patients with AIS treated with MT using clinically relevant preoperative and postoperative time point variables have not been developed. Currently, most are built based on preoperative variables, such as THRIVE [10], HIAT [14], GADIS [19], NAC [20], and IER-START [21], and several machine learning (ML) models [22,23,24,25], and these scores and models cannot be updated according to the changes in the patient’s state and examination results over time. In clinical practice, these scores are intended to inform treatment, and they lack focus on determining functional outcomes after treatment. Additionally, the above scores have been validated externally, and the AUC range was from 0.680 to 0.838 in recent studies [10,20,21], indicating that there is still room to improve accuracy. In addition, these scores, based on the linear regression algorithm, have some limitations in addressing nonlinear problems between the variables in real-world applications. Therefore, developing a high accuracy model for a clinically relevant admission, postoperative 24-h, 3-day, with discharge time points for the dynamical prediction of unfavorable outcomes is desirable. 

ML algorithms are in the realm of artificial intelligence. They have been proven to be an excellent approach in modeling multi-factor events in medical fields, such as brain–computer interface (BCI) technology [26,27,28], deep brain stimulation (DBS) application [29], and outcome prediction [30,31], because ML can contain a mass of data, extract subtle information, and summarize the knowledge gained on new unseen situations efficiently and automatically. At the same time, ML can also deal with the non-linear relationship between data. As can be surmised, applying an ML model at multiple time points after admission has allowed neurologists to better dynamically assess functional neurological outcomes and adjust therapeutic strategies. 

To improve personalized stroke care, we developed and validated a dynamic ML model at multiple time points for the dynamic prediction of a three-month unfavorable outcome. 

## 2. Materials and Methods

### 2.1. Study Population

We analyzed electronic health record data from three Chinese stroke units (Nanjing First Hospital, Changsha Central Hospital, and People’s Hospital of Hunan Province) of adult patients (≥18 years of age at the time of surgery) between January 2014 and December 2018. Patients included in the analysis went through the following: (i) had an AIS treated with MT; (ii) had their three-month modified Rankin Scale (mRS) scores assessed; (iii) their National Institute of Health stroke scale (NIHSS) scores on admission, postoperative 24-h, 3-day, and discharge were recorded. Patients aged <18 years or with an intracranial hemorrhage (ICH) on admission or interval from onset to treatment over 24 h were excluded.

In this study, the candidate variables for our model included demographics (age, sex, smoking), medical history (previous cerebral infarction, transient ischemic attack, previous cerebral hemorrhage, diabetes mellitus, hyperlipidemia, hypertension, atrial fibrillation and coronary heart disease), baseline data (diastolic pressure, systolic pressure, NIHSS score on admission, Trial of ORG 10172 in Acute Stroke Treatment (TOAST) classification, anterior circulation stroke, posterior circulation stroke, and blood biochemical examination), as well as interventional characteristics (interval from groin puncture to recanalization, interval from onset to treatment, endovascular therapy and IV thrombolysis) and post interventional clinical characteristics (symptomatic intracranial hemorrhage (sICH), NIHSS score after 24-h, NIHSS score after 3-day and NIHSS score on discharge). 

We estimated that the outcome was neurological–functional three months after the AIS with MT, as represented by an mRS score >2 (i.e., poor prognosis). In addition, the symptomatic intracranial hemorrhage (sICH) was exposed within 24 h after admission and evaluated by the Heidelberg Bleeding Classification [32]. The NIHSS score and three-month mRS were collected by outpatient services and telephone interviews with patients or their families.

### 2.2. Statistical Analyses

Variables with less than 20% missing values were taken in to generate the complete dataset. As the case may be, missing variables were imputed by mode for categorical variables and mean or median for continuous variables

A statistical analysis was performed using SPSS version 22.0 (IBM Corporation, Armonk, NY, USA), applying the number (percentage) as a measure of categorical variables and the medians (quartile) or means (standard deviation) as measures of continuous variables, as appropriate. In addition, comparisons in the baseline characteristics between favorable outcome and unfavorable outcome groups for continuous variables were performed using the Mann–Whitney U test or Student t test, depending on their normality of distribution, and Pearson’s test or Fisher’s exact test for categorical variables, depending on their sample amount.

### 2.3. Feature Selection

Feature selection further reduced redundancy and irrelevant variables, which is a critical step in improving a model’s stability. Concerning feature selection, we ranked the model features consistent with their importance (importance type = “gain,” the equal gain of divides that apply to the feature) and chose a group of features by eliminating the least important features. To avoid the false-negative result in the univariable analysis, all features with *p* < 0.2 in the univariable analysis or that were considered to have clinical significance features were taken into the feature selection. The features selected from the model could be divided into four subsets, judging by the time at which they were collected: admission, postoperative 24-h, postoperative 3-day, and discharge.

### 2.4. Model Development

Predictive ML models for the three-month outcome of AIS patients treated were developed by eXtreme gradient boosting (XGBoost). XGBoost is an ensemble learning classifier that utilizes a Gradient Boosting framework to solve supervised learning problems. XGBoost has been widely used in medical studies to predict or screen prognosis [33,34,35]. XGBoost also uses a trained prediction model to provide a regularized gradient enhancement and feature importance score, which can be applied to feature selection. For model development, it was performed in Python (version 3.7). We built machine-learning models with four sets of input variables corresponding to time points, and it was named the “Admission” model, “24-Hour” model, “3-Day” model, and “Discharge” model, respectively. The following procedure was carried out in each model: First, patients were randomly split into a train set and a test set according to an 8:2 scale procedure. Then, the train data were used to train and build the model. Meanwhile, we utilized a grid search algorithm and ten-fold cross-validation to optimize model hyperparameters. In a ten-fold cross-validation, the training set was randomly divided into ten equal folds, each fold was used in turn for the validation set, and the remaining nine folds were used for the training set, which improved the generalization of the model and avoided overfitting. By obtaining the models, the model’s performance was measured on the testing set. 

### 2.5. Model Evaluation

We used the area under the receiver operating characteristic curve (AUC) to measure discrimination [36]. The calibration of the models was assessed using the Brier scores method (range: 0–1)—a model with a lower Brier score means better model calibration and discrimination [37]. The AUC comparisons between different models were evaluated with the DeLong test [38]. We further applied the testing set to THRIVE [10] and HIAT [14] scores to evaluate the dynamic model’s preponderance in predictive performance.

## 3. Results

### 3.1. Study Population

The clinical characteristics of the unfavorable and favorable outcome groups are shown in Table 1. We screened 239 AIS patients treated with MT, patients for age <18 (*n* = 1; 0.4%) were excluded, and our final study sample included 238 patients overall, within the follow-up period, with a median age of 69 (IQR 59–78), and 72 patients (30.3%) were women. The median admission NIHSS score was 15 (IQR 10–20), while the median NIHSS score at postoperative 24-hour, 3-day, and discharge decreased to 13 (IQR 6–21), 12 (IQR 4–23), and 9 (IQR 2–20), respectively. Of the 238 patients, 156 (cumulative rate 65.5%) patients developed unfavorable outcomes in three months. and 61 patients (cumulative rate 25.6%) died (mRS scores = 6). The clinical outcome assessed by mRS scores at three months was 0 in 24 (10%), 1 in 25 (10.5%), 2 in 33 (13.8%), 3 in 34 (14.2%), 4 in 39 (16.3%), 5 in 24 (10%), and (10.5%), 2 in 33 (13.8%), 3 in 34 (14.2%), and 4 in 6 in 59 patients (24.7%). Compared with favorable outcome patients, patients with unfavorable outcomes were seven years older and had lower low-density lipoprotein, lower platelets, and higher creatinine. Patients with an unfavorable outcome also had longer intervals from groin puncture to recanalization and a higher NIHSS score.

### 3.2. Feature Selection

According to the results (Appendix A), we selected the top 10 variables that had a higher importance from the feature selection results as input variables of an “Admission” model. Similarly, the input variables of three other models that were the same as before the NIHSS score and were consistent with the time point were the highest-ranked feature in all models.

### 3.3. Model Performance

From the identified top-performing XGBoost-selected predictors, the prediction models for the three-month outcome of AIS patients treated with MT were created. Figure 1a,b presents the ROC curves of four models on the training set and testing set, respectively. Table 2, Table 3, Table 4 and Table 5 present the confusion matrix of the performance of the “Admission”, “24-Hour”, “3-Day”, and “Discharge” models on the testing dataset. In addition, the Brier scores for unfavorable functional outcome prediction AIS on the testing dataset are shown in Figure 2.

#### 3.3.1. “Admission” Model

On the training data, the “Admission” model achieved an AUC of 0.835. On the testing set, the model achieved an AUC of 0.824, and the sensitivity and specificity were 0.848 and 0.533. Among the 33 cases with an unfavorable outcome, 28 cases were correctly predicted by the model. The model incorrectly predicted five cases as favorable outcomes, leading to a recall rate of 70%. A total of 75.0% of cases were correctly predicted, contributing to an error rate of 25.0%. The hyper-parameters applied in the “Admission” model are shown in Appendix A.

#### 3.3.2. “24-Hour” Model

On the training data, the “24-Hour” model achieved an AUC of 0.917. On the testing set, the model achieved an AUC of 0.891, and the sensitivity and specificity were 0.788 and 0.800. Among the 33 cases with an unfavorable outcome, 26 cases were correctly predicted by the model. The model incorrectly predicted seven cases as favorable outcomes, leading to a recall rate of 78.8%. A total of 79.2 % of cases were correctly predicted, corresponding to an error rate of 14.6%. The hyper-parameters applied in the “24-Hour” model are shown in Appendix A.

#### 3.3.3. The “3-Day” Model

On the training data, the “3-Day” model achieved an AUC of 0.937. On the testing set, the model achieved an AUC of 0.931, and the sensitivity and specificity were 0.758 and 0.933. Among the 33 cases with an unfavorable outcome, 25 cases were correctly predicted by the model. The model incorrectly predicted eight cases as favorable outcomes, leading to a recall rate of 75.8%. A total of 81.2% of cases were correctly predicted, corresponding to an error rate of 18.8%. The hyper-parameters applied in the “3-Day” model are shown in Appendix A.

#### 3.3.4. “Discharge” Model

On the training data, the “Discharge” model achieved an AUC of 0.987. On the testing set, the model achieved an AUC of 0.945, and the sensitivity and specificity were 0.909 and 0.800. Among the 33 cases with an unfavorable outcome, 30 cases were correctly predicted by the model. The model incorrectly predicted three cases as favorable outcomes, leading to a recall rate of 90.9%. A total of 87.5% of cases were correctly predicted, corresponding to an error rate of 12.5%. The hyper-parameters applied in the “Discharge” model are shown in Appendix A.

**Table 2 brainsci-12-00938-t002:** Confusion matrix for the “Admission” Model.

Testing Data				Statistical Analysis	
True Predicted	0	1	Total	Accuracy	0.750
0	8	7	15	Precision	0.800
1	5	28	33	Sensitivity	0.848
Total	13	35	48	Specificity	0.533
				AUC	0.824

AUC, the area under curve.

**Table 3 brainsci-12-00938-t003:** Confusion matrix for the “24-H” Model.

Testing Data				Statistical Analysis	
True Predicted	0	1	Total	Accuracy	0.792
0	12	3	15	Precision	0.897
1	7	26	33	Sensitivity	0.788
Total	19	29	48	Specificity	0.800
				AUC	0.891

AUC, the area under curve.

**Table 4 brainsci-12-00938-t004:** Confusion matrix for the “3-Day” Model.

Testing Data				Statistical Analysis	
True Predicted	0	1	Total	Accuracy	0.812
0	14	1	15	Precision	0.962
1	8	25	33	Sensitivity	0.758
Total	22	26	48	Specificity	0.933
				AUC	0.931

AUC, the area under curve.

**Table 5 brainsci-12-00938-t005:** Confusion matrix for the “Discharge” Model.

Testing Data				Statistical Analysis	
True Predicted	0	1	Total	Accuracy	0.875
0	12	3	15	Precision	0.909
1	3	30	33	Sensitivity	0.909
Total	17	31	48	Specificity	0.800
				AUC	0.945

### 3.4. Model Comparison

Among all prediction models, we found that the “Discharge” model had the highest AUC (0.970; 95%Cl 0.901–0.991) and the lowest Brier scores (0.081), and the “Admission” model had the lowest AUC (0.824; 95%Cl 0.715–0.899) and the highest Brier scores (0.122). The results of the DeLong test indicated that the “Admission” and “24-Hour” models were statistically different from the AUC of “Discharge” models (*p* < 0.05) (Appendix A), and the AUC of the four models gradually increased from admission to discharge on the testing set. Meanwhile, the Brier scores decreased from admission to discharge on the testing set. The dynamic model exhibited a good predictive ability on the testing set and improved over time. In addition, the AUCs of the dynamic model were significantly superior to THRIVE [10] and HIAT [14] scores (Figure 1b, Appendix A).

## 4. Discussion

To the best of our knowledge, this is the first attempt to construct four XGBoost ML models (i.e., a preoperative, postoperative 24-h, 3-day, and discharge model) to dynamically predict the probability of a three-month unfavorable outcome in Chinese patients with AIS treated with MT. Although traditional prediction scores [10,14,19,20,21] and ML models [22,23,24,25] offer practical guidance and are effective stroke prognosis tools for population-based comparisons, they predict the outcome of AIS patients undergoing MT by a static nature, and their clinical use is limited. Instead, in our study, we combined preoperative and intraoperative data with time-relevant postoperative variables to dynamically predict the three-month outcome of MT for AIS patients. the “Admission” model could be used to predict the clinical outcome before MT and support the decision to perform MT, and the “24-Hour”, “3-Day”, and “Discharge” models would further improve the predictive accuracy of the clinical outcome after MT and adjust therapeutic strategies in time. From admission to discharge, the physiology or pathology will change along with treatment, so it is necessary to consider all possible factors that occurred until a specific time point and provide an outcome prediction in real time that could guide better decisions for treatment [39]. With that in mind, we proposed a method of risk-based dynamic monitoring strategies and targeted the introduction of therapy neurologists when discussing prognosis with AIS patients and their families.

One strength of our study lies in our models showing good discrimination ability compared to previous scoring systems (THRIVE and HIAT scores) by the AUC of models at different time points. In the “Admission” model, it included preoperative factors, and the AUC of the model was 0.824, while in the “24-Hour”, “3-Day”, and “Discharge” models, they included intraoperative and postoperative factors, and the AUC of the model was 0.891–0.945, while THRIVE and HIAT scores with an AUC of 0.653 and 0.658, respectively. The recent single-center cohort study combined individual data of 246 patients from Heidelberg University Hospital with multiple clinical characteristics to develop unfavorable outcome prediction models [23]. The baseline model included clinical and conventional imaging characteristics on admission to reach an AUC of 0.740 to predict unfavorable outcomes in internal validation. Further addition of CT-perfusion, angiographic, and post-interventional characteristics achieved an AUC of 0.856, still below the predictive performance of our model [23]. These four prediction models are beneficial to reducing the risk of postoperative three-month unfavorable outcome and may have good application prospects. In addition, however, we also found that adding intraoperative data with time-relevant postoperative variables (i.e., NIHSS score in postoperative 24-h, 3-day, and discharge time points) would modestly improve the predictive performance of outcome models, which supported the theory that the admission NIHSS scores are inferior to postoperative 24-h NIHSS, 3-day NIHSS, and discharge NIHSS score in predicting post-stroke functional outcome [40]. 

There were several additional strong points of this study. First, we derived unfavorable outcome models using state-of-the-art ML methods, which have recently been applied in the development of neurological function outcome prediction models in the domain of neurological disease [22,23,24,25]. However, our modeling strategy differed from the conventional single modeling strategy, with multiple models predicting the unfavorable outcome for more precise forecasting. Second, the prediction models were developed using multicenter registration data. A wide range of data sources may reduce non-sampling errors and enhance the reliability of models. Third, it is notable that, in most studies, the importance of the features were first assessed using univariate analysis and based on prior knowledge, then the significant features were engaged in the ML model. In the end, they calculated the variable importance of variables in ML models. This variable selection method is called the filter method and has been widely used. However, it may introduce too many variables to fit models well and may bring the collinearity of variables [23]. Thus, the rank of feature importance in the model was introduced to feature selection, reducing over-fitting and avoiding variables collinearity.

In our study, the variables selected were inexpensive to obtain and commonly available. Therefore, we found it to be similar to former prediction models. The most important features among the four XGBoost models were the NIHSS score, which achieved a dominant position in all variables, and confirmed NIHSS was relatively high in the three-month outcome of AIS patients [40]. Furthermore, a previous study said the NIHSS score after endovascular surgery had a better discrimination in predicting three-month functional outcomes than the pretreatment NIHSS score [41]. Therefore, the “24-Hour”, “3-Day”, and “Discharge” models, which incorporated both preoperative NIHSS score and postoperative NIHSS score could be offered modest improvements in predicting a three-month unfavorable outcome. 

It is consistent with previous reports, which incorporated the discharge NIHSS score with good predictive accuracy about the functional outcome. For example, Miguel Monteiro et al. [42] constructed machine learning prognostic models for AIS patients, and the models revealed that incorporating the discharge NIHSS score had the best result in predicting the 90-day functional outcome. Nevertheless, in this paper, all patients were treated using Recombinant Tissue Plasminogen Activator (rtPA), and the literature included many variables, which was not conducive to predicting clinical promotion. Furthermore, our models not only contributed to accurately predicting patients who would and would not benefit from MT, but also they would benefit from improving the postoperative accuracy of AIS patients with MT.

Nevertheless, our study has several limitations. First, this study was based on a retrospective data set, and there may be invisible disturbing factors contributing to potential biases in the results. Thus, prospective studies are necessary and validated in an independent external cohort to reconfirm these results in future work. Second, the dynamic models screened out patients with poor outcomes but did not suggest who would benefit from preventative therapy. It still depends on clinicians to decide whether to use therapeutic interventions. However, as the disease continues, clinical indicators may change. The early and precise prediction of poor outcomes may require more time for neurologists to reform treatment strategies and reduce complications and mortality. Third, the size of our study population and clinical characteristics were limited, while current ML models lack an underlying causal structure: their predictions are entirely based on what humans have done in the past. Hence, the future will expand the sample size for further research.

## 5. Conclusions

This is the first dynamic predictive model of a three-month unfavorable outcome for Chinese AIS patients with MT that was developed and validated at admission, postoperative 24-h, 3-day, and discharge time points, and these models showed good discrimination and calibration. In terms of AUCs, the range can be up to 0.945, depending on when the prediction was made. Furthermore, compared with previous scoring (THRIVE and HIAT scores), the dynamic risk prediction had a better performance. The preoperative model could be used to predict the clinical outcome before MT and support the decision to perform MT, and the postoperative models would further improve the predictive accuracy of the clinical outcome after MT and adjust therapeutic strategies in timely. 

## Figures and Tables

**Figure 1 brainsci-12-00938-f001:**
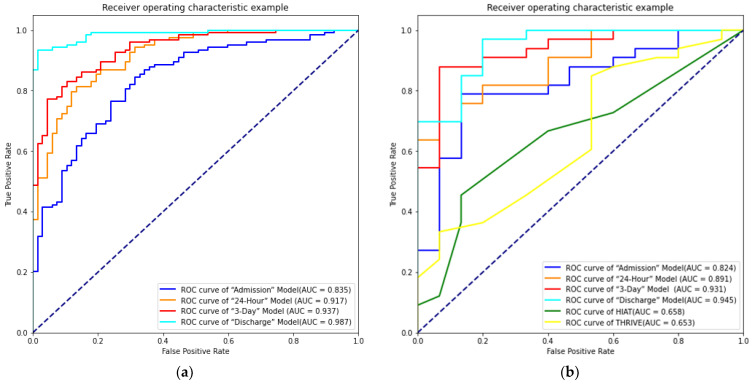
(**a**) The receiver operating characteristic curve (ROC) of four models on the training set; (**b**) the receiver operating characteristic curve (ROC) of four models and traditional scores on testing set. AUC, the area under curve; THRIVE, Totaled Health Risks in Vascular Events; HIAT, Houston Intra-arterial Recanalization Therapy.

**Figure 2 brainsci-12-00938-f002:**
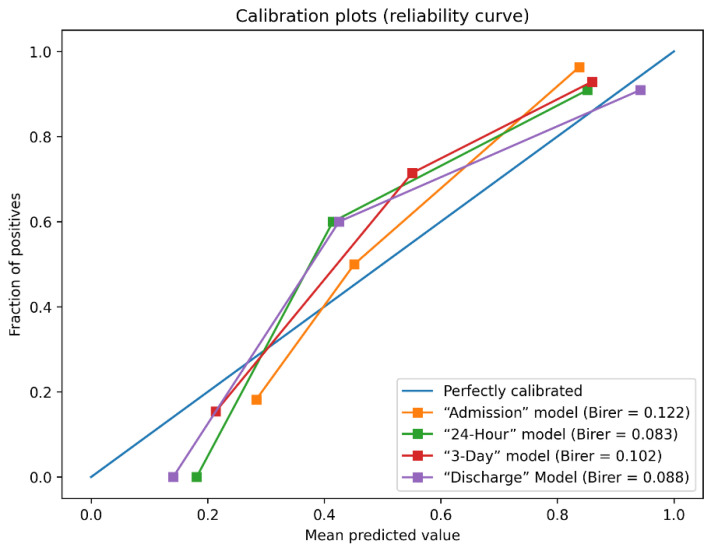
The calibration curves and the Brier score of four models on the testing set.

**Table 1 brainsci-12-00938-t001:** Clinical, demographic and laboratory data of study population stratified according to three-month favorable or unfavorable outcome after acute ischemic stroke in Chinese patients with mechanical thrombectomy.

Variables	Favorable Outcome (mRS 0–2)	Unfavorable Outcome (mRS 3–6)	*p*-Value
Patients, *n*	82	156	
**Demographics**			
Age, years, median (IQR)	65 (58–74)	72 (60–81)	0.001 *
Sex, *n* (%)			0.592
Male	59 (72)	107 (68.6)	
Female	23 (28)	49 (31.4)	
Smoking, *n* (%)	28 (34.1)	49 (31.4)	0.668
**Medical history, *n* (%)**			
Transient ischemic attack	0 (0)	3 (1.9)	0.110 *
Previous cerebral infarction	12 (14.6)	29 (18.6)	0.443
Previous cerebral hemorrhage	1 (1.2)	6 (3.8)	0.462
Diabetes mellitus	18 (22)	31 (19.9)	0.706
Hypertension	57 (69.5)	108 (69.5)	0.964
Hyperlipidemia	7 (8.5)	12 (7.7)	0.819
Coronary artery disease	15 (18.3)	44 (28.2)	0.092 *
Atrial fibrillation	25 (30.5)	58 (37.2)	0.303
**Baseline data**			
NIHSS score on admission, median (IQR)	11 (7–16)	17 (13–22)	<0.0001 *
Systolic pressure, mmHg, median (IQR)	138 (124–155)	142 (129–160)	0.299
Diastolic pressure, mmHg, median (IQR)	83 (74–93)	86 (76–99)	0.154 *
INR, median (IQR)	0.98 (0.93–1.09)	1.02 (0.935–1.12)	0.092 *
HbA1c, mmol/L, median (IQR)	5.80 (5.50–6.53)	5.90 (5.50–6.50)	0.856
TC, mmol/L, median (IQR)	4.32 (3.43–4.98)	4.08 (3.43–4.83)	0.300
TG, mmol/L, median (IQR)	1.12 (0.82–1.68)	1.05 (0.76–1.47)	0.205
LDL, mmol/L, median (IQR)	2.66 (1.98–3.24)	2.23 (1.80–2.83)	0.019 *
FBG, mmol/L, median (IQR)	6.04 (5.08–7.35)	6.48 (5.60–7.99)	0.028 *
PLT, μmol/L, median (IQR)	193.00 (150.75–235.50)	172.5 (143.00–212.50)	0.017 *
UA, μmol/L, median (IQR)	284.90 (233.00–357.00)	313.50 (232.32–396.75)	0.125 *
HCY, μmol/L, median (IQR)	12.46 (10.70–16.76)	13.18 (10.97–16.64)	0.433
Creatinine, μmol/L, median (IQR)	67.00 (58.37–77.00)	78.00 (61.63–94.00)	0.003 *
Anterior circulation stroke, *n* (%)	60 (73.2)	126 (80.8)	0.178 *
Posterior circulation stroke, *n* (%)	22 (26.8)	30 (19.2)	0.178 *
TOAST classification, *n* (%)			0.119 *
Large artery atherosclerosis	47 (57.3)	69 (44.2)	
Cardioembolism	32 (39.0)	75 (48.1)	
Others	3 (3.7)	12 (7.7)	
**Interventional characteristics**			
Interval from groin puncture to recanalization, min, median (IQR)	60 (50–85)	81 (59–130)	0.004 *
Interval from onset to treatment, min, median (IQR)	290 (230–411)	280 (206–413)	0.240
Endovascular therapy, *n* (%)			0.144 *
Tirofiban	29 (35.4)	41 (26.3)	
No tirofiban	53 (64.6)	115 (73.7)	
IV thrombolysis, *n* (%)			0.806
No thrombolysis	45 (54.9)	83 (53.2)	
Thrombolysis	37 (45.1)	73 (46.8)	
**Post interventional characteristics**			
sICH, *n* (%)	0 (0)	18 (11.5)	0.001 *
NIHSS score after 24-hour, median (IQR)	5 (2–10)	17 (12–31)	<0.0001 *
NIHSS score after 3-day, median (IQR)	3 (2–7)	18 (10–34)	<0.0001 *
NIHSS score on discharge, median (IQR)	2 (1–3)	16 (8–34)	<0.0001 *

* Included into the feature selection (*p* < 0.2). mRS, modified Rankin Scale; NIHSS, National Institute of Health stroke scale; TOAST, Trial of ORG 10172 in Acute Stroke Treatment; INR, International normalized ratios; IQR, interquartile range; HbAc1, Glycated hemoglobin; TC, total cholesterol; TG, triglyceride; LDL, Low density lipoprotein; FBG, Fasting blood glucose; PLT, Platelet; UA, Uric Acid; HCY, Homocysteine; IV, Intravenous; sICH, symptomatic intracranial hemorrhage.

## Data Availability

The datasets we used in this study are available from the corresponding author upon reasonable request.

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
