# Peer review of "Dynamic Prediction of Mechanical Thrombectomy Outcome for Acute Ischemic Stroke Patients Using Machine Learning"

_brainsci, 2022, doi:10.3390/brainsci12070938_

Round 1
Reviewer 1 Report
The authors propose a machine learning approach for dynamic prediction of mechanical Thrombectomy outcome for Acute Ischemic Stroke. In general, the main conclusions presented in the paper are supported by the figures and supporting text. However, to meet the journal quality standards, the following comments need to be addressed.
1. Abstract: Should be improved and extended. The authors talk lot about the problem formulation, but novelty of the proposed model is missing. Also provided the general applicability of their model. Please be specific what are the main quantitative results to attract general audiences.
2. The introduction can be improved. The authors should focus on extending the novelty of the current study. Emphasize should be given in improvement of the model (in quantitative sense) compared to existing state-of-the art models.
3. More details about network architecture and complexity of the model should be provided.
4. what about comparison of the result with current state-of-the art models? Did authors perform ablation study to compare with different models?
5. What are the baseline models and benchmark results? The authors can compared the result with existing models evaluated with datasets
6. Conclusion parts needs to be strengthened.
7. Please provide a fair weakness and limitation of the model, and how it can be improved.
8. Typographical errors: There are several minor grammatical errors and incorrect sentence structures. Please run this through a spell checker.
9. Following references can be added as deep learning references for brain signal classification (see Bio. Sig. Pro. Cont. 74, (2022) , 103101 https://doi.org/10.1016/j.bspc.2021.103101
Int. J. Human–Computer Interaction 38, 2022 https://doi.org/10.1080/10447318.2021.1921482,
Bioxiv 2022 https://doi.org/10.1101/2022.03.17.481909).
Author Response
Question 1. Abstract: Should be improved and extended. The authors talk lot about the problem formulation, but novelty of the proposed model is missing. Also provided the general applicability of their model. Please be specific what are the main quantitative results to attract general audiences.
Author’s responses: Thank you very much for good suggestions.
We have added novelty in the new abstract and provided the general applicability of model and made it specific what are the main quantitative results to attract general audiences. Which are is marked in bold and underline as following:
Abstract: The unfavorable outcome of acute ischemic stroke (AIS) with large vessel occlusion (LVO) is related to clinical factors at multiple time points. However, predictive models for dynamically predicting unfavorable outcomes using clinically relevant preoperative and postoperative time point variables have not been developed. Our goal was to develop a machine learning (ML) model for dynamic prediction of unfavorable outcomes. We retrospectively reviewed patients with AIS who underwent consecutive MT from three centers in China between January 2014 and December 2018. Based on XGBoost algorithm, we used clinical characteristics on admission (“Admission” Model), and additional variables regarding intraoperative management and postoperative National Institute of Health stroke scale (NIHSS) score (“24-Hour” Model, “3-Day” Model and “Discharge” Model). The outcome was the unfavorable outcome at three-month (modified Rankin scale, mRS 3-6: unfavorable). The area under the receiver operating characteristic curve and Brier scores were the main evaluating indexes. Unfavorable outcome at three-month was observed in 156 (62.0%) of 238 patients. These four models had high accuracy in the range of 75.0% to 87.5% and had good discrimination with AUC in the range of 0.824 to 0.945 on the testing set. The Brier scores of the four models ranged from 0.122 to 0.083 and showed good predictive ability on the testing set. This is the first dynamic, preoperative and postoperative predictive model constructed for AIS patients who underwent MT, which show more accurately than previously prediction model. The preoperative model could be used to predict the clinical outcome before MT and support the decision to perform MT, and the postoperative models would further improve the predictive accuracy of clinical outcome after MT and adjust therapeutic strategies in timely.
Question 2. The introduction can be improved. The authors should focus on extending the novelty of the current study. Emphasize should be given in improvement of the model (in quantitative sense) compared to existing state-of-the art models.
Author’s responses: Thank you very much for good suggestions.
(1) We have added the novelty of the current research, and which is marked in bold and underline as following:
The models for dynamically predicting three-month unfavorable outcomes in patients with AIS treated with MT using clinically relevant preoperative and postoperative time point variables have not been developed. Currently, most predictive models are built based on preoperative variables, such as THRIVE [10], HIAT [14], GADIS [19], NAC [20] and IER-START [21] and several machine learning (ML) models [22-25], these scores and models cannot be updated according to the changes in the patient’s state and examination results over time. Although, in clinical practice, these scores are intended to inform treatment, and lack of focus on determining functional outcomes after treatment. Additionally, the above scores have been validated externally and the AUC range from 0.680 to 0.838 in recent studies [10,20,21], indicating there is still room to improve accuracy. Besides, these scores based on the linear regression algorithm have some limitations in addressing nonlinear problems between the variables in real-world applications. Therefore, developing a high accuracy model at clinically relevant admission, postoperative 24-hour, 3-day and discharge time points for dynamical prediction of unfavorable outcomes is desirable.
ML algorithms are in the realm of artificial intelligence. They have been proven to be an excellent approach in modeling multi-factor events in medical fields, such as Brain-Computer Interface (BCI) technology [26-28], Deep brain stimulation (DBS) application [29]and outcome prediction [30,31], due to ML can contain a mass of data, extract subtle information, and summarize the knowledge gained on new unseen situations efficiently and automatically. At the same time, ML can also deal with the non-linear relationship between data. As can be surmised, applying ML model at multiple time points after admission has allowed neurologists to better dynamic assess functional neurological outcome and adjust therapeutic strategies.
(2) We have emphasized the improvements to the model (in quantitative sense) compared to existing state-of-the art models, and which is marked in bold and underline as following:
The models for dynamically predicting three-month unfavorable outcomes in patients with AIS treated with MT using clinically relevant preoperative and postoperative time point variables have not been developed. Currently, most predictive models are built based on preoperative variables, such as THRIVE [10], HIAT [14], GADIS [19], NAC [20] and IER-START [21] and several machine learning (ML) models [22-25], these scores and models cannot be updated according to the changes in the patient’s state and examination results over time. Although, in clinical practice, these scores are intended to inform treatment, and lack of focus on determining functional outcomes after treatment. Additionally, the above scores have been validated externally and the AUC range from 0.680 to 0.838 in recent studies [10,20,21], indicating there is still room to improve accuracy. Besides, these scores based on the linear regression algorithm have some limitations in addressing nonlinear problems between the variables in real-world applications. Therefore, developing a high accuracy model at clinically relevant admission, postoperative 24-hour, 3-day and discharge time points for dynamical prediction of unfavorable outcomes is desirable.
ML algorithms are in the realm of artificial intelligence. They have been proven to be an excellent approach in modeling multi-factor events in medical fields, such as Brain-Computer Interface (BCI) technology [26-28], Deep brain stimulation (DBS) application [29]and outcome prediction [30,31], due to ML can contain a mass of data, extract subtle information, and summarize the knowledge gained on new unseen situations efficiently and automatically. At the same time, ML can also deal with the non-linear relationship between data. As can be surmised, applying ML model at multiple time points after admission has allowed neurologists to better dynamic assess functional neurological outcome and adjust therapeutic strategies.
Question 3. More details about network architecture and complexity of the model should be provided.
Author’s responses: Thank you very much for good suggestions.
We have added more details about the network architecture and complexity to the article, which is listed as following (the key points are marked in bold and underline):
Feature selection: We ranked the model featured consistent with their importance. (importance type = “gain,” the equal gain of divides that apply the feature) and chose a group of features by eliminating the least important feature. To avoid the false-negative result in the univariable analysis, all features with P < 0.2 in the univariable analysis or considered to have clinical significance features were taken into feature selection.
Model development: The following procedure was carried out in each model: First, patients were randomly split into a train set and a test set according to an 8:2 scale procedure. Then the train data was used to train and build the model. Meanwhile, we utilized a grid search algorithm and ten-fold cross-validation to optimize model hyperparameters. In ten-fold cross-validation, the training set was randomly divided into ten equal folds, each fold was used in turn to the validation set, and the remaining nine folds were used for the training set, which improved the generalization of the model and avoided overfitting. With obtaining the models, the model's performance was measured on the testing set.
Model evaluation: We used the area under the receiver operating characteristic curve (AUC) to measure discrimination [35].Calibration of models was assessed using the Brier scores method (range: 0–1)—a model with a lower Brier score means bet-ter model calibration and discrimination [36]. The AUC comparison between different models were evaluated with DeLong test [37]. We further applied the testing set to THRIVE [10] and HIAT [14] scores to evaluate the dynamic model's preponderance in predictive performance.
Question 4. what about comparison of the result with current state-of-the art models? Did authors perform ablation study to compare with different models?
Author’s responses: Thank you very much for good suggestions.
(1) The results were compared to the current state-of-the art models and which is marked in bold and underline as following:
In recent studies, we found a single-center cohort study combined individual data of 246 patients from Heidelberg University Hospital of multiple clinical characteristics and develop several unfavorable outcome predication models [1], which is current state-of the art models. The baseline model included clinical and conventional imaging characteristics on admission to reach an AUC of 0.740 to predict unfavorable outcome in internal validation. Further addition of CT-perfusion, angiographic and post interventional characteristics achieving an AUC of 0.856. While Our model had good discrimination with AUC in the range of 0.824 to 0.945 better than the current state-of-the art models.
(2) Did authors perform ablation study to compare with different models?
I’m sorry that we didn’t perform ablation study in this study. Because we build dynamic model based on eXtreme gradient boosting (XGBoost) algorithm. XGBoost offers the ability to tune and optimize a range of hyperparameters, i.e. model parameters that are not derived from underlying data but can be set to control the learning process. Additionally, XGBoost has an advantage in that it is computationally efficient, handles missing data effectively, and provides both a probability output as well as insight into the importance of each parameter [2].
References:
1. Nishi H, Oishi N, Ishii A et al. Predicting Clinical Outcomes of Large Vessel Occlusion Before Mechanical Thrombectomy Using Machine Learning. Stroke. 2019;50(9):2379-2388
2. Zea-Vera R, Ryan CT, Havelka J et al. Machine Learning to Predict Outcomes and Cost by Phase of Care After Coronary Artery Bypass Grafting. Ann Thorac Surg. 2021;
Question 5. What are the baseline models and benchmark results? The authors can compared the result with existing models evaluated with datasets.
Author’s responses: Thank you very much for good suggestions.
(1) The baseline models and benchmark results as following:
We introduced THRIVE [1] and HIAT [2] scores as baseline models and its AUC were regarded as benchmark results.
(2) We have added the comparison of dynamic prediction model with baseline models as following:
The AUCs of the dynamic model were significantly superior to THRIVE[1] and HIAT[2] scores.(Figure1, Supplemental Table 2).
Supplemental TABLE 2. The p-values of pairwise comparisons of AUCs on the testing set for different models with the Delong test |
|||||
Model |
“24H” model |
“3Days” model |
“Discharge” model |
THRIVE |
HIAT |
“Admission” model |
0.011 | 0.081 | 0.015 | 0.003 | 0.012 |
“24H” model |
0.235 | 0.021 | 0.001 | 0.008 | |
“3Days” model |
0.139 | 0.002 |
0.003 |
||
“Discharge” model |
|
|
|
0.001 |
0.002 |
THRIVE |
|
|
|
|
0.950 |
The significant difference between AUCs is defined as p-value < 0.05. Abbreviations: AUC, the area under receiver operating characteristic curve; THRIVE, Totaled Health Risks in Vascular Events; HIAT, Houston Intra-arterial Recanalization Therapy.
|
References:
1. Flint AC, Cullen SP, Faigeles BS, Rao VA. Predicting long-term outcome after endovascular stroke treatment: the to-taled health risks in vascular events score. AJNR Am J Neuroradiol. 2010;31(7):1192-6
2. Hallevi H, Barreto AD, Liebeskind DS et al. Identifying patients at high risk for poor outcome after intra-arterial therapy for acute ischemic stroke. Stroke. 2009;40(5):1780-5
Queetion 6. Conclusion parts needs to be strengthened.
Author’s responses: Thank you very much for good suggestions. We have strengthened Conclusion parts as following (the key points are marked in bold and underline):
This is the first dynamic predictive model of three-month unfavorable outcome for Chinese AIS patients with MT was developed and validated at admission, postoperative 24-hour, 3-day and discharge time points, and these models show good discrimination and calibration. In terms of AUCs the range can be up to 0.945, depending on when the prediction was made. Furthermore, compared with previous scoring (THRIVE and HIAT scores), the dynamic risk prediction had a better performance. And the preoperative model could be used to predict the clinical outcome before MT and support the decision to perform MT, and the postoperative models would further improve the predictive accuracy of clinical outcome after MT and adjust therapeutic strategies in timely.
Question 7. Please provide a fair weakness and limitation of the model, and how it can be improved.
Author’s responses: Thank you very much for good suggestions. We have added a fair weakness and limitation of the model, and how it can be improved as following:
First, this study was based on a retrospective data set, and there may be invisible disturbing factors that might contribute to potential biases in the results. So prospective studies are necessary and validated in an independent external cohort to reconfirm these results in the future work. Second, the dynamic models screened out patients with poor outcome but do not suggest who will benefit from preventative therapy. It still depends on clinicians to decide whether to therapeutic interventions. However, as the disease continues, clinical indicators may change. Early and precise prediction of poor outcome may strive more time for neurologists to reform treatment strategies, and reduce complications and mortality. Third, the size of our study population and clinical characteristics were limited, while current ML models lack underlying causal structure: their predictions are entirely based on what humans have done in the past. Hence, future will expand the sample size for further research.
Question 8. Typographical errors: There are several minor grammatical errors and incorrect sentence structures. Please run this through a spell checker.
Author’s responses: Thanks for your constructive suggestion, which is highly appreciated.
We have scrutinized the manuscript, and made corresponding revisions including some contraction, grammatical errors and long sentences, etc. In addition, the expression of the manuscript has been improved with the help of a native English speaker.
Question 9. Following references can be added as deep learning references for brain signal classification.
Author’s responses: Thanks for your constructive suggestion.
We read the article you recommended carefully. Using brain-computer interface (BCI) to establish direct communication between the brain and external electronic devices is a great study. The research is in the field of artificial intelligence. Therefore, the literature mentioned by the reviewer has been incorporated into the revised manuscript.
We have cited the literature in our article: ML algorithms are in the realm of artificial intelligence. They have been proven to be an excellent approach in modeling multi-factor events in medical fields, such as Brain-Computer Interface (BCI) technology [1-3].
Reference:
1. Roy AM. A multi-scale fusion CNN model based on adaptive transfer learning for multi-class MI-classification in BCI system. BioRxiv. 2022;
2. Sharma R, Kim M, Gupta A. Motor imagery classification in brain-machine interface with machine learning algorithms: Classical approach to multi-layer perceptron model. Biomed Signal Proces. 2022;71:103101
3. Kapgate D. Efficient Quadcopter Flight Control Using Hybrid SSVEP + P300 Visual Brain Computer Interface. Int J Hum-Comput Int. 2022;38(1):42-52

Reviewer 2 Report
The introduction should provide more information about ischemic stroke, main symptoms, risk factors...
It would be advisable to include more relevant references in this field
Great care must be taken when extrapolating and generalizing the results of this study.
On the other hand, the sample is small.
It would be advisable to review the English of this manuscript
Author Response
Comments and Suggestions for Authors:
Question 1. The introduction should provide more information about ischemic stroke, main symptoms, risk factors...
Author’s responses: Thank you very much for good suggestions.
In the introduction, we have provided more information about ischemic stroke and added more relevant references in this field. In addition, the outcome of stroke patients is determined by the patient's clinical presentation [1]. However, we described the modified Rankin Scale (mRS) score in detail here (Table1), because of the limitation of the words.
Previous studies show that many prognostic factors are associate with the unfavorable outcome in AIS patients. Such as physiologic factors [10-12] (e.g., age, sex, comorbidity), clinical factors [13,14] (e.g., NHISS score, glycosylated hemoglobin, creatinine) and neuroimaging prognostic factors [15-17] (e.g., location of the occlusion, Alberta Stroke Program Early CT Score (ASPECTS)). However, these pretreatment prognostic factors still cannot well predict the outcome of AIS patients. It is due to some interventional factors also affects the prognosis, like procedure time, surgical technique and recanalization status and so on [10,14,18].
Table 1. The Modified Rankin Scale
Grade |
Description |
0 |
No symptoms |
1 |
Minor symptoms that do not interfere with lifestyle |
2 |
Minor handicap, symptoms that lead to some restriction in lifestyle but do not interfere with the patient's capacity to look after himself |
3 |
Moderate handicap, symptoms that clearly prevent independent existence |
4 |
Moderately severe handicap, symptoms that clearly prevent independent existence though not needing constant attention |
5 |
Severe handicap, totally dependent patient requiring constant attention night and day |
6 |
Death |
Reference:
1. Wu S, Wu B, Liu M et al. Stroke in China: advances and challenges in epidemiology, prevention, and management. Lancet Neurol. 2019;18(4):394-405
2. Flint AC, Cullen SP, Faigeles BS, Rao VA. Predicting long-term outcome after endovascular stroke treatment: the to-taled health risks in vascular events score. AJNR Am J Neuroradiol. 2010;31(7):1192-6
3. Almekhlafi MA, Davalos A, Bonafe A et al. Impact of age and baseline NIHSS scores on clinical outcomes in the me-chanical thrombectomy using solitaire FR in acute ischemic stroke study. AJNR Am J Neuroradiol. 2014;35(7):1337-40
4. Hametner C, Kellert L, Ringleb PA. Impact of sex in stroke thrombolysis: a coarsened exact matching study. Bmc Neu-rol. 2015;15:10
5. Linfante I, Starosciak AK, Walker GR et al. Predictors of poor outcome despite recanalization: a multiple regression analysis of the NASA registry. J Neurointerv Surg. 2016;8(3):224-9
6. Hallevi H, Barreto AD, Liebeskind DS et al. Identifying patients at high risk for poor outcome after intra-arterial thera-py for acute ischemic stroke. Stroke. 2009;40(5):1780-5
7. Liggins JT, Yoo AJ, Mishra NK et al. A score based on age and DWI volume predicts poor outcome following endovas-cular treatment for acute ischemic stroke. Int J Stroke. 2015;10(5):705-9
8. Lansberg MG, Christensen S, Kemp S et al. Computed tomographic perfusion to Predict Response to Recanalization in ischemic stroke. Ann Neurol. 2017;81(6):849-856
9. Elijovich L, Goyal N, Mainali S et al. CTA collateral score predicts infarct volume and clinical outcome after endovas-cular therapy for acute ischemic stroke: a retrospective chart review. J Neurointerv Surg. 2016;8(6):559-62
10. Rangaraju S, Liggins JT, Aghaebrahim A et al. Pittsburgh outcomes after stroke thrombectomy score predicts out-comes after endovascular therapy for anterior circulation large vessel occlusions. Stroke. 2014;45(8):2298-304
Question 2. It would be advisable to include more relevant references in this field.
Author’s responses: Thank you very much for good suggestions.
Based on Question1, we have included more relevant references in the field of stroke and presented in Question1’s reference.
Question 3. Great care must be taken when extrapolating and generalizing the results of this study.
Author’s responses: Thank you very much for good suggestions.
The Article was comprehensively reviewed and make our result extrapolating and generalizing be more care cautious.
Although our model gives better results, this study was based on a retrospective data set, and there may be invisible disturbing factors that might contribute to potential biases in the results. So prospective studies are necessary and validated in an independent external cohort to reconfirm these results in the future work. And the dynamic models screened out patients with poor outcome but do not suggest who will benefit from preventative therapy.
Question 4. On the other hand, the sample is small.
Author’s responses: Thank you very much for good suggestions.
The number of Chinese AIS patients with MT is very small. Because the proportion of AIS patients who underwent MT treatment in patients may be less 1.74% [1], which needs to be further improved. In AIS patients, Mechanical thrombectomy (MT) is now regarded as a standard of care for the management of patients with acute large vessel occlusion stroke [2-5]. In other studies of AIS patients who underwent MT, such as “Multimodal Predictive Modeling of Endovascular Treatment Outcome for Acute Ischemic Stroke Using Machine-Learning[6]”, “Machine Learning for Outcome Prediction of Acute Ischemic Stroke Post Intra-Arterial Therapy[7]” and “Predicting Clinical Outcomes of Large Vessel Occlusion Before Mechanical Thrombectomy Using Machine Learning[8]”, specific patients included in these studies for training and validation ML models were 246, 107, 387, respectively.
1. 霍晓川, 李晓青, 缪中荣, 国家神经系统疾病医疗质量控制中心神经介入质控专家委员会, 急性脑梗死再灌注治疗质量改进国家行动血管内治疗工作委员会. 2020中国急性缺血性卒中血管内治疗现状调查分析[J]. 中国卒中杂志, 2022, 17(01): 56-65.
2. Saver JL, Goyal M, van der Lugt A et al. Time to Treatment With Endovascular Thrombectomy and Outcomes From Ischemic Stroke: A Meta-analysis. JAMA. 2016;316(12):1279-88
3. Schonewille WJ. BEST evidence on mechanical thrombectomy for patients with vertebrobasilar occlusion. Lancet Neurol. 2020;19(2):102-103
4. Powers WJ, Rabinstein AA, Ackerson T et al. 2018 Guidelines for the Early Management of Patients With Acute Ischemic Stroke: A Guideline for Healthcare Professionals From the American Heart Association/American Stroke .Association. Stroke. 2018;49(3):e46-e110
5. Jovin TG, Chamorro A, Cobo E et al. Thrombectomy within 8 hours after symptom onset in ischemic stroke. N Engl J Med. 2015;372(24):2296-306
6. van Os H, Ramos LA, Hilbert A et al. Predicting Outcome of Endovascular Treatment for Acute Ischemic Stroke: Potential Value of Machine Learning Algorithms. Front Neurol. 2018;9:784
7. Asadi H, Dowling R, Yan B, Mitchell P. Machine learning for outcome prediction of acute ischemic stroke post intra-arterial therapy. Plos One. 2014;9(2):e88225
8. Predicting Clinical Outcomes of Large Vessel Occlusion Before Mechanical Thrombectomy Using Machine Learning. Stroke. 2019;50(9):2379-2388
Question 5. It would be advisable to review the English of this manuscript.
Author’s responses: Thanks for your constructive suggestion, which is highly appreciated. We have carefully scrutinized the manuscript, and made corresponding revisions including some contraction, grammatical errors and long sentences, etc. In addition, the expression of the manuscript has been improved with the help of a native English speaker.

Reviewer 3 Report
Excellent paper, very original work of the authors who developed a machine learning model who can predict three-month unfavorable outcome for Chinese AIS patients with MT. Very comprehensive and well organized work; the methods used, the tables and images used are very explicit.
I don't see any issues not to be published in current form. Excellent!
Author Response
Comments and Suggestions for Authors
Excellent paper, very original work of the authors who developed a machine learning model who can predict three-month unfavorable outcome for Chinese AIS patients with MT. Very comprehensive and well-organized work; the methods used, the tables and images used are very explicit. I don't see any issues not to be published in current form. Excellent!
Author’s responses: Thank you very much for positive comment. We have made further improvements to our article during the revision time.

Round 2
Reviewer 1 Report
The current manuscript is suitable for publication.
Reviewer 2 Report
Good improvement in your paper, I congratulate you.
I hope you continue to investigate this topic.